# Assessment of Antimicrobial Activity, Mode of Action and Volatile Compounds of *Etlingera pavieana* Essential Oil

**DOI:** 10.3390/molecules25143245

**Published:** 2020-07-16

**Authors:** Porawan Naksang, Sasitorn Tongchitpakdee, Kanjana Thumanu, Maria Jose Oruna-Concha, Keshavan Niranjan, Chitsiri Rachtanapun

**Affiliations:** 1Department of Food Science and Technology, Faculty of Agro-Industry, Kasetsart University, Bangkok 10900, Thailand; porawan.n@ku.th (P.N.); sasitorn.ch@ku.th (S.T.); 2Synchrotron Light Research Institute (Public Organization), Nakhon Ratchasima 30000, Thailand; kanjanat@slri.or.th; 3Department of Food and Nutritional Sciences, University of Reading, Whiteknights, Reading RG6 6AP, UK; m.j.oruna-concha@reading.ac.uk (M.J.O.-C.); afsniran@reading.ac.uk (K.N.); 4Center for Advanced Studied Agriculture and Food, Kasetsart University, Bangkok 10900, Thailand

**Keywords:** *Etlingera pavieana*, essential oil, antimicrobial activity, foodborne pathogens, mode of action

## Abstract

*Etlingera pavieana* (Pierre ex Gagnep.) R.M.S. is a rhizomatous plant in the Zingiberaceae family which could be freshly eaten, used as a condiment or as a traditional remedy. Our work investigated the chemical composition and antimicrobial activity of the *E. pavieana* essential oils extracted from the rhizome (EOEP). We extracted the EOEP from the rhizome by hydrodistillation and analyzed the chemical composition by headspace solid-phase microextraction coupled with gas chromatography/mass spectrometry (HS-SPME-GC/MS). A total of 22 volatile compounds were identified where *trans*-anethole (78.54%) and estragole (19.36%) were the major components in the EOEP. The antimicrobial activity of EOEP was evaluated based on the minimum inhibitory concentration (MIC) and the minimum bactericidal concentration (MBC) values using the broth dilution method and enumerating cell death overtime. Our work shows that the EOEP exhibits potent antibacterial activity against foodborne pathogenic Gram-positive bacteria, namely *Bacillus cereus*, *Staphylococcus aureus* and *Listeria monocytogenes* in the range of 0.1–0.3% (*v*/*v*). We further investigated the mechanism of EOEP inhibition using Synchrotron Fourier transformation infrared (FTIR) microspectroscopy. Here, we show significant differences in DNA/nucleic acid, proteins and cell membrane composition in the bacterial cell. To conclude, EOEP exhibited antimicrobial activity against foodborne pathogens, especially the Gram-positive bacteria associated with ready-to-eat (RTE) food and, thus, has the potential to serve as a natural preservative agent in RTE products.

## 1. Introduction

*Etlingera pavieana* (Pierre ex Gagnep.) R.M.S., Raew-hawm in Thai, is an aromatic plant in the genus *Etlingera* belonging to the Zingiberaceae, or the ginger family. It is widely distributed in south-eastern Thailand and the Cardamon Mountains in Cambodia. The rhizome of *E. pavieana*, as shown in Figure 1, has been used as a traditional medicine to heal symptoms, such as nausea, fever and flatulence. It is also used as a diuretic [1,2]. In a local Thai food, known as Moo Lieng noodle, which is a traditional dish in Chantaburi and Trad province, it is used as the main ingredient. There is some research evidence to demonstrate its antimicrobial activity against *Mycobacterium tuberculosis*, *Escherichia coli*, *Staphylococcus aureus*, *Bacillus cereus* and *Listeria monocytogenes*, and it has also been shown to possess antioxidant, anti-inflammatory and anticancer activities in human breasts [2,3,4,5]. However, its bioactivities are largely varied, depending upon factors, such as plant parts, extraction method employed and the solvent used [4].

A number of recent studies on the volatile composition of *Etlingera* spp. focus on the essential oil obtained either by steam distillation or hydrodistillation, as well as others by basic extraction techniques involving the use of organic solvents. However, there are relatively few studies on the volatile compounds of Zingiberaceae extracts obtained using alternative methods, such as headspace solid-phase microextraction coupled with gas chromatography/mass spectrometry (HS-SPME-GC/MS. The HS-SPME-GC/MS is increasingly used for fast and automatic extraction coupled with a high-throughput analysis of a large number of volatile compounds in a single step without involving the use of solvents [6,7]. Despite many traditional uses and medicinal claims of different species of *Etlingera*, very few scientific studies have focused on the essential oil. Particularly, the mode of action of the essential oils was much less mentioned. As far as we know, no information is available on the investigation into these antimicrobial mechanisms using Synchrotron FTIR microspectroscopy. Thus, this research aimed to extract the volatile compounds from *E. pavieana* analyze the antimicrobial activity of the extract and investigate the mechanism of its action against pathogenic bacteria present in ready-to-eat (RTE) products so that the extract can be used as a natural food preservative.

## 2. Results and Discussion

### 2.1. Yield Percentage and Analysis of the Volatile Compounds from the Rhizome of E. pavieana Essential Oils (EOEP)

The EOEP obtained after hydrodistillation shows a light yellow color. The yield percentage of EOEP was 0.13% (*v*/*w* on fresh weight basis), which is slightly lower than the products obtained from other rhizomes of *Etlingera* spp., which range between 0.28 and 0.70% (*v*/*w* on dry weight basis) [2,8]. However, this is much greater than the estimated oil yield on a fresh weight basis, which is at 0.019–0.047% (*v*/*w*) [9].

The volatile composition of the EOEP from rhizome is shown in Table 1. Twenty-two compounds have been identified, representing 99.94% of the total oil composition. This amount comprised 2 phenylpropanoids (97.90%), 11 monoterpenes (1.55%), 7 oxygenated monoterpenes (0.40%) and 2 sesquiterpenes (0.09%), respectively. *Trans*-anethole (78.54%) and methyl chavicol or estragole (19.36%) were the major compounds (Figure 2), followed by β-myrcene (0.84%), camphene (0.17%), pinocarvone (0.15%), camphor (0.12%) and α-pinene (0.12%), respectively. This result is in accordance with Ud-Daula and Basher [10], who reported that phenylpropanoids (non-terpenic compounds) were the main chemical group in the rhizome essential oils of the *E. pavieana*, *Etlingera brevilabrum* (Valeton) R.M.Sm., *Etlingera cevuga* (Seem.) R.M.Sm., *Etlingera linguiforme* (Roxb.) R.M.Sm., *Etlingera littoralis* (J. Koenig) Giseke and *Etlingera punicea* (Roxb.) R.M.Sm. Among the phenylproponoids, *trans*-anethole (48.6%) was the major compounds in the essential oil of *E. pavieana*, and methyl chavicol dominated the rhizome oil of *E. punicea* (95.75%) and *E. linguiforme* (49.9%), respectively [2,11,12]. In the work of Tachai [2], who analyzed the hydrodistilled essential oil from dried rhizomes of *E. pavieana* in the Chantaburi province, Thailand, trans-anethole (48.6%), *p*-anisaldehyde (13.8%), δ-cadinene (7.2%), *trans*-methyl isoeugenol (2.7%) and α-cadinol (2.4%) were the major components. Furthermore, a series of phenylpropanoids, such as 4-methoxycinnamaldehyde, 4-methoxycinnamyl *p*-coumarate, 4-methoxycinnamyl alcohol and *p*-coumaric acid, have been extracted from the rhizome of *E. pavieana* from Thailand [3,5]. In addition, terpenes (monoterpene hydrocarbons and sesquiterpene hydrocarbons) and terpenoids (oxygenated monoterpenes and oxygenated sesquiterpenes) were found in small proportions in the essential oil of *E. pavieana* [2,10]. However, those terpinic compounds are abundant in most *Etlingera* species, including *E. brevilabrum, Etlingera elatior* (Jack) R.M.Sm., *Etlingera fimbriobracteata* (K.Schum.) R.M.Sm., *E. megalocheilos* (Griff.) A.D. Poulsen, comb. nov. and *Etlingera sayapensis* A.D. Poulsen and Ibrahim [8,13,14,15,16]. Monoterpene hydrocarbons, such as α-pinene, camphene, β-myrcene, d-Limonene and β-ocimene, were found in high percentages in the essential oil, in particular, in the oils from leaves [10,12,13]. Comparing the components of essential oils from *Etlingera* species, the chemical compositions are influenced by the plant species and parts, sources and the methods of preparation [13,17].

### 2.2. Determination of the MIC and MBC of EOEP

The antimicrobial activity of EOEP from the rhizome against foodborne microorganisms was assessed by MIC and MBC values investigated by broth dilution assay, as shown in Table 2. The results show that EOEP has a stronger antimicrobial activity against Gram-positive bacteria than Gram-negative bacteria. The MIC and MBC values against Gram-positive bacteria range between 0.02 to 0.03% (*v*/*v*) and 0.10 to 0.30% (*v*/*v*), respectively, while the MIC and MBC values against most Gram-negative bacteria and lactic acid bacteria are 5% (*v*/*v*) or greater. Thus, this EOEP is not strongly effective against Gram-negative bacteria. Amongst the bacteria tested, *L. monocytogenes* strain 108 and Scott A were found to be most sensitive strains with the MIC and MBC values of 0.02% (*v*/*v*) and 0.10% (*v*/*v*), respectively, followed by *B. cereus*, which gave values of 0.02% and 0.15% (*v*/*v*), respectively. The antimicrobial activity of the *E. pavieana* essential oil has been reported in few studies. There was only one report on the biological activities of the *E. pavieana* essential oil and, although it showed cytotoxic activity on human small-cell lung cancer (NCI-H187) cells with an IC_50_ value of 31.69 μg/mL, it was inactive against *Mycobacterium tuberculosis* [18]. From our previous publication, various solvent extracts of *E. pavieana,* including ethanol, acetone, dichloromethane, ethyl acetate, petroleum ether and hexane, had antibacterial activity against Gram-positive bacteria *(Bacillus cereus*, *B. subtilis*, *Staphylococcus aureus* and *Listeria monocytogenes*) and Gram-negative bacteria (*Escherichia coli*, *Pseudomonas aeruginosa*, *Vibrio parahaemolyticus* and *Salmonella typhimurium*) [4]. Comparing the antimicrobial activity of the *Etlingera* essential oils, the essential oils from *E. elatior*, *Etlingera fulgens*, *Etlingera maingayi, E. punicea* and *Etlingera rubrostriata* had antibacterial activity against Gram-positive bacteria, namely *B. cereus, Micrococcus luteus* and *S. aureus* [11,19,20]. Moreover, the essential oil of *E. elatior* flowers and *E. punicea* had antifungal activity against *Candida albicans* [11,20]. However, those essential oils were ineffective against Gram-negative bacteria, including *E. coli*, *Pseudomonas aeruginosa, Salmonella albany* and *Salmonella* Choleraesuis. These reports correspond to the findings we have presented above. It has been generally reported that Gram-negative bacteria are more resistant to antimicrobial agents due to the lipopolysaccharide layer on the outer membrane of Gram-negative bacteria being highly hydrophilic. This acts as a strong barrier against hydrophobic molecules, which are the majority of the antimicrobial agents. In contrast, the cell walls of the Gram-positive bacteria only consist of thick peptidoglycan, which make the bacteria more susceptible to attack by the antibacterial agents [13,20,21,22]. Interestingly, lactic acid bacteria, which are mostly Gram-positive, are more resistant to EO than the other Gram-positive members [23,24,25,26]. This effect may be due to the surface of the lactic acid bacteria, including *Lactobacillus* spp., which are able to provide intrinsic resistance to some antibacterial agents [23].

### 2.3. Enumeration Cell Death Over Time by Time-Kill Curve

Time-kill assay of EOEP against the tested bacterial species was performed to determine the response on the growth pattern. Based on the MIC and MBC values, six concentrations of EOEP were tested against foodborne pathogenic Gram-positive bacteria, including *B. cereus*, *S. aureus* and *L. monocytogenes,* at an initial concentration of ca. 3 log CFU/mL. The bacterial viable cells were enumerated over time to investigate their response to the antimicrobial agent, as shown in Figure 3. The bacteriostatic (the concentration where the bacterial culture remains constant) and bactericidal (the concentration where the bacteria dies at no viable count in 18h) concentrations can be determined from Figure 3 [27]. All bacterial strains grew well in the control sample (non-selective media without the addition of EOEP). The cell numbers of *B. cereus* and *S. aureus* in the control (TSB without EOEP) reached 9 Log CFU/mL in 12 h, as shown in Figure 3A,B. When 0.01% EOEP was added to the broth medium, *B. cereus* cells grew to 9 Log CFU/mL in 12 h, which was similar to the control. In 0.03% EOEP, the bacterial cell count remained constant at ca. 2 log CFU/mL for 12 h and then increased to 8 Log CFU/mL after 18 h. On the other hand, bactericidal effect was detected at an EOEP concentration of 0.05% as the number of cells gradually reduced to undetectable levels within 18 h. Furthermore, at a concentration of 0.15%, the number of *B. cereus* cells became undetectable within 12 h. The response of *S. aureus* to 0.01 and 0.03% EOEP was similar to *B. cereus*, as shown in Figure 3B. In the case where 0.05% EOEP was applied, it showed a bacteriostatic effect, as seen in the fact that the cell numbers remained constant at ca. 2 log CFU/mL throughout the study. At 0.10 and 0.15%, EOEP became bactericidal, and the number of cells reduced to undetectable amounts within 24 and 18 h, respectively. The response of *L. monocytogenes* to EOEP was more sensitive than other species, as it showed bacteriostatic effect at 0.03% EOEP, and became bactericidal at 0.05% EOEP. The *listerial* cell was completely inactivated at 0.10% EOEP within 24 h, as shown in Figure 3C. In summary, time-kill curves revealed the dynamic response of the bacterial cells on the growth pattern at an EOEP concentration of 0.01%, which showed an inhibitory effect against *B. cereus*, *S. aureus* and *L*. *monocytogenes* Scott A. At a concentration of 0.03%, EOEP showed bacteriostatic effect and, at 0.1% or greater, EOEP showed lethal effect.

Though reports on the antimicrobial activity of EOEP are limited, its inhibitory activity against test bacteria could be attributed to the presence of bioactive compounds, mainly *trans*-anethole and methyl chavicol (estragole), as previously reported [11,21,28,29,30]. *Trans*-Anethole at a concentration of 75 μg/mL showed clear inhibition of the growth of *S. aureus* at inoculum level of 10^5^ cells/plate [28]. Estragole had a strong inhibitory effect against *Pseudomonas syringae* in the liquid culture assay throughout 24 h. The effectiveness of estragole was stronger than eugenol > methyl-eugenol > cinnamaldehyde, respectively [29]. Moreover, the antibacterial activity of estragole is comparatively higher than that of *trans*-anethole [30]. The estragole-rich oil exhibited MICs of 3.7, 6.7 and 13.2 mg/mL against *C. albican*, *S. aureus* and *E. coli,* respectively, while *trans*-anethole-rich oil showed a MIC of 1.8 mg/mL against *C. albican*. In contrast, the antioxidative activity of estragole is comparatively less than *trans*-anethole [31]. Nonetheless, it has been shown to carry antioxidative activity. Both *trans*-anethole and methyl chavicol (estragole) have shown both antioxidant and antimicrobial activities, but very little is known about their modes of action against target microorganisms.

### 2.4. Determination of the Modes of Action of EOEP on B. cereus Using Synchrotron FTIR Microspectroscopy

#### 2.4.1. Assessment of Antimicrobial Action of EOEP Using Synchrotron FTIR Microspectroscopy

The effect of EOEP on the cellular components of *B. cereus* was investigated by Synchrotron FTIR microspectroscopy. FTIR spectra of cells treated with 0.1% EOEP for 0, 6 and 12 h are shown in Figure 4A. The FTIR spectra for most bacteria were divided into four regions comprising the fatty acid in the cell (Region I; peak from 3000 to 2800 cm^−1^), the amide groups from proteins and peptides (Region II; peak from 1700–1500 cm^−1^), the mixed region governed by fatty acids, proteins and phosphate-carrying molecules (Region III; peak from 1500–1200 cm^−1^) and nucleic acid, DNA and RNA (Region IV; peak from 1200–1000 cm^−1^) [32]. The spectra obtained for *B. cereus* corresponded to those previously described [33,34]. Strong absorption units obtained for all regions, demonstrating the main components of a cell. To assess the detailed structure of the FTIR spectra and characterize the related variations among bacterial FTIR spectra, the second derivative was analyzed, as shown in Figure 4B,C. The representative FTIR band assignments of biomolecule groups are shown in Table 3.

The FTIR micrographs revealed the change in cellular components in the spectral intensity at the dominant peaks at 3295, 2925, 2851, 1657, 1541, 1444, 1400, 1240 and 1085 cm^−1^. Overall, the cells show significant changes after being treated with EOEP for 6–12 h (*p* ≤ 0.05). After cells were treated for 6 h, the most apparent change occurs at the DNA/nucleic acid region where the absorptions of bands 1240 and 1085 cm^−1^ increase about 4- and 3-fold, respectively. The band at 1240 cm^−1^ represented the phosphate group (P=O) asymmetric stretching of phosphodiesters in phospholipids and RNA [35,36] and 1085 cm^−1^ represented a phosphate symmetric stretching in DNA, RNA and phospholipids [36,37]. We also observed a 2-fold increase in the Amide I of α-helical structures (1650 cm^−1^), the Amide I of β-pleated sheet structures (1630 cm^−1^) and the Amide II (1541 cm^−1^) region after 6 and 12 h of treatment, respectively [35,37]. Moreover, we hypothesized a major disruption in the cell membrane due to the change in the two absorbance bands at 1400 cm^−1^ and 1444 cm^−1^, with the former attributed to the symmetric stretching of the COO-group in amino acids and fatty acids in the cell membrane [36] and the deformation of the CH_2_ of lipids and fatty acids in the cell membrane [35]. The increase in the spectra at 1650, 1541, 1444, 1400, 1240 and 1085 cm^−1^ suggests a leakage of cellular components. These changes generally reflect changes in the cell membrane and the leakage of the nucleic acids into the solvent. These results showed the most substantial cellular biomolecule changes in phosphodiesters in nucleic acid, including DNA and RNA, followed by cell membrane and proteinaceous substances after being treated with EOEP for 6 h.

#### 2.4.2. Principal Component Analysis (PCA) of the FTIR Spectra from *B. cereus* Treated with EOEP

We conducted PCA to justify this multivariate data set and differentiate changes between untreated cells and treated cells. Here, we showed clear data segregations between untreated cells and treated cells of the scores for the PC1 (variability; 60%) and PC2 (variability; 23%), as shown in Figure 5. The PCA score plot shows that spectra from a cell treated with 0.1% EOEP for 0, 6 and 12 h on *B. cereus* can be clustered separately along PC1 and PC2. Moreover, a grouping of the spectra represents significant differences between untreated (0 min) and treated cells (6 and 12 h) that can be correlated with changes in the molecular composition of *B. cereus*, including fatty acid (2925 cm^−1^), protein (1630 cm^−1^), and mixed region, which characterizes the cell membrane and DNA/nucleic acids (1240 and 1085 cm^−1^). The PCA results correspond to the time-kill assay, as previously described, as the EOEP at concentration 0.1% showed bactericidal effect against *B. cereus* within 6 h and continuously throughout 12 h. Interestingly, the negative and positive spectral variations indicated by the PC1 and PC2 loading plots that were oppositely correlated to each other at 1630 cm^−1^ may indicate a higher level of amide I of β-pleated sheet structures in the proteinaceous content of treated cells compared to untreated cells. Additionally, the oppositely correlated PC1 and PC2 at 1240 and 1085 cm^−1^ may reflect considerable level of nucleic acids, DNA and RNA of the treated and untreated cells. Our findings reveal the mode of action of the EOEP targeted on the phosphodiester of the nucleic acids, including DNA and RNA and the proteinaceous substances of cells, in particular, amide I.

## 3. Materials and Methods

### 3.1. Plant Material

The *E. pavieana* was cultivated in a commercial local farm in the Trad province, Thailand. The plant was collected and identified by an expert herbalist. The whole plant of *E. pavieana* was harvested in August 2015. The fresh rhizomes of *E. pavieana* were washed to remove soil and chopped into small pieces. The chopped rhizomes (100 kg) were fed into a vertical hydrodistillation machine (HVAC Engineering Co. Ltd., Pathumthani, Thailand) for 10 h. The essential oil was stored at −30 °C in an airtight container for further use. The total amount of essential oil was recorded. The yield percentage was calculated on a fresh weight basis.

### 3.2. Characterization of Chemical Composition of EOEP

The volatile compositions of EOEP from the rhizome were characterized by a headspace solid-phase microextraction system (HS-SPME) using a 50/30 μm divinylbenzene (DVB)/ polydimethylsiloxane (PDMS) fiber (Supelco, Bellefonte, PA, USA), according to the method of Morales-Soto et al. [5] and Saoudi et al. [6] with some modifications. The EOEP, 0.5 g, was weighed and 4.5 mL of ethyl ether was added to a SMPE vial of 15 mL fitted with a screw cap. After equilibration at 40 °C for 10 min, the fiber was exposed to the headspace above the sample for 30 min. The sample was kept under stirring at 40 °C and desorbed for 20 min in the GC injector at 250 °C. Analysis was done in triplicate. The GC-MS analysis of SPME extracts was carried out as described by Morales-Soto et al. [5]. Briefly, the volatile compounds were identified by comparison with mass spectra from NIST/EPA/NIH databases and confirmed in many cases by the comparison of their retention indices with databases (http://www.pherobase.com) and the ADAMS library. To confirm the identification, the linear retention index (LRI) was calculated for each volatile, using the retention times of a homologous series of C6–C25 *n*-alkanes. Semiquantitative determinations were expressed as the percentage of total peak area within each sample.

### 3.3. Microorganisms Preparation

In total, we tested 17 strains of foodborne microorganisms. These include Gram-positive bacteria, namely *B. cereus* TISTR687, *S. aureus* TISTR1466, *L. monocytogenes* strain 101, 108, V7 and Scott A, lactic acid bacteria, namely *L. pentosus* TISTR920, *L. plantarum* strains TISTR541, TISTR844 and TISTR850, *L. mesenteroides* TISTR053 and *P. cerevisiae* and, finally, Gram-negative bacteria, namely *E. coli* TISTR780, *S. enterica* serotype Typhimurium ATCC13311 and *P*. *aeruginosa* TISTR781. For the culture preparation, *B. cereus*, *S. aureus*, *S.* Typhimurium and *P. aeruginosa* were grown in TSB at 37 °C, except for *B. cereus*, which was cultured at 30 °C. *Listeria* spp. was grown in TSB containing 0.6% yeast extract (YE) at 37 °C and lactic acid bacteria were grown in MRS broth. The culture was then subjected to two successive 24 h and 18 h transfers before use.

### 3.4. Determination of Antimicrobial Activity

#### 3.4.1. MIC and MBC Assay

The broth dilution method was used for determining both the MIC and MBC of the EOEP against all strains of tested bacteria. This method was used as modified from Sukmark et al. [46]. Test tubes containing 5 mL of each bacterium (4–5 Log CFU/mL) and double strength Mueller–Hinton broth (MHB; Merck, Darmstadt, Germany), MHBYE for *Listeria* spp. or MRS broth for lactic acid bacteria were mixed to a single concentration in either EOEP or sterilized deionized (DI) water to give a final volume of 10 mL. The EOEP was mixed into a test tube to give a final concentration in the range of 0.01 to 5% (*v*/*v*). The methodology also included positive and negative controls. The positive control included tubes containing inoculum and MHB medium without EOEP, and the negative control comprised tubes containing EOEP and MHB medium, without inoculum. The test tubes with only MHB medium served as a blank control. After mixing, the test tubes were incubated at 37 °C, except *B. cereus* was incubated at 30 °C for 24 h. Survival numbers of tested bacteria were examined using the spread plate method. *B. cereus*, *S. aureus*, *E. coli*, *P. aeruginosa* and *S.* Typhimurium were grown on TSA. *Listeria* spp. was grown on TSAYE and lactic acid bacteria was grown in MRS agar. Then, the agar was incubated at 37 °C, while *B. cereus* was incubated at 30 °C. All experiments were repeated in triplicate and duplicated in each experiment.

The MIC was defined as the lowest concentration of antimicrobial concentration required to inhibit the visible growth in the test tube. The MBC was defined as the lowest concentration of antimicrobial concentration required to diminish more than 99.9% (3 Log reduction) of the initial bacterial amount at 24 h [46,47].

#### 3.4.2. Time-kill Assay

The antimicrobial activity of EOEP over time was determined against *B. cereus*, *S. aureus* and *L. monocytogenes* Scott A. This method was modified from Thongson, et al. [48] and Ekkarin, et al. [49]. *B. cereus* and *S. aureus* were grown in TSB and *L. monocytogenes* Scott A was grown in TSBYE for 18 h and serially diluted into 10 mL double strength broth media, which was contained in a 125 mL Erlenmeyer flask. Then, a certain amount of EOEP was added to maintain the concentration at 0 (a positive control), 0.01, 0.03, 0.05, 0.1 and 0.15% (*v*/*v*). Sterile distilled water was then added to adjust the total volume to 20 mL. The final concentration of the initial cell was 3 Log CFU/mL. All flasks were incubated at 37 °C, except *B. cereus*, which was incubated at 30 °C in a shaker at 100 rpm for up to 24 h. Numbers of survivor were taken at 0, 1, 3, 6, 9, 12, 18 and 24 h and viable cells were enumerated on agar media. All experiments were repeated in triplicate and duplicated in a single repeat.

### 3.5. Determination of Mode of Actions of EOEP on B. cereus Using Synchrotron FTIR Microspectroscopy

#### 3.5.1. Synchrotron FTIR Sample Preparation

*B. cereus* cells were grown at 30 °C in TSB medium added with EOEP. The samples were taken immediately after contact with EOEP (0 h), and then after 6 h and 12 h. A 5 μl aliquot of *B. cereus* was transferred into a new sterile microtube and centrifuged at 12,000 rpm for 2 min. A pellet was washed twice using sterile 0.85% NaCl and DI water. The suspended culture was placed on an IR-transparent 2-mm-thick barium fluoride (BaF_2_) slide. Samples on the slide were dried at room temperature for 2 h under laminar flow for dehydration. The dried samples on the BaF_2_ slides were kept in a vacuum and dried before FTIR analysis. The Synchrotron FTIR microspectroscopy was performed at Beamline 4.1 in the Synchrotron Light Research Institute (SLRI, Nakhon Ratchasima, Thailand). The FTIR microspectra were obtained from the wavenumber range of 3000–800 cm^−1^. In total, 270 spectra (90 spectra per replicate) were acquired at room temperature. The measurement was performed in the mapping mode using an aperture size 10 × 10 μm with a spectral resolution of 6 cm^−1^ and 64 scans co-added. Spectral acquisition and instrument control were performed using an OPUS 6.5 (Bruker, Ettlingen, Germany).

#### 3.5.2. FTIR Data Analysis

Before all spectra of *B. cereus* cells were collected each time, background was collected each time. All spectra were nine-point smoothed, normalized, baseline corrected and integrated using the OPUS 6.5 software. A second derivative of the spectrum was calculated using the Savitzky–Golay algorithm and subjected to the multivariate statistical technique of principal component analysis (PCA) by Unscramble X software (Camo Analytics, Oslo, Norway).

### 3.6. Statistical Analysis

The data obtained in this study are expressed as the mean of at least three replicates determined and standard deviation (S.D.). Analysis of variance (ANOVA) was performed using Duncan multiple comparison tests at *p* ≤ 0.05 by SPSS version 12.0 (SPSS Inc., Chicago, IL, USA) for Windows.

## 4. Conclusions

Essential oils obtained from the *E. pavieana* rhizome contain high *trans*-anethole and estragole content, allowing them to inhibit Gram-positive bacteria associated with ready-to-eat (RTE) foods, namely *L. monocytogenes, B. cereus* and *S. aureus*. The main active compounds reacted mostly with the cell plasma membrane, causing the DNA/nucleic acid and protein to leak. This study provides the benefit of using EOEP as a natural food preservative in certain liquid foods system.

## Figures and Tables

**Figure 1 molecules-25-03245-f001:**
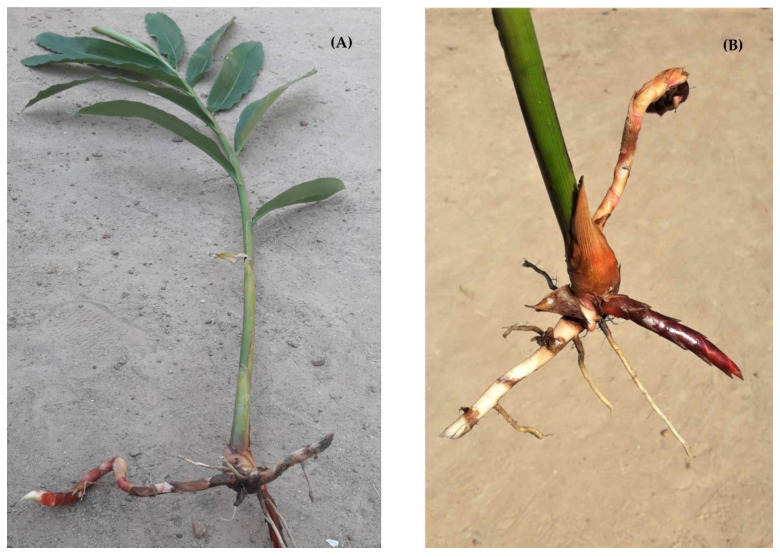
Photographs of *Etlingera pavieana* (Pierre ex Gagnep.) R.M.Sm. plant (**A**) and *E. pavieana* rhizome (**B**).

**Figure 2 molecules-25-03245-f002:**
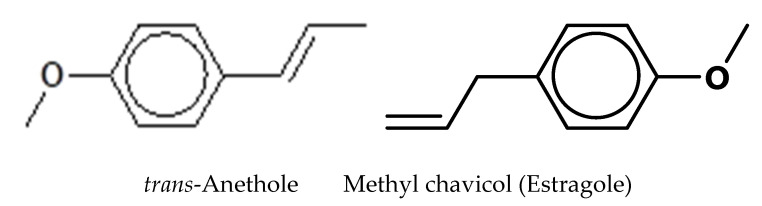
Structures of the main compounds identified in EOEP.

**Figure 3 molecules-25-03245-f003:**
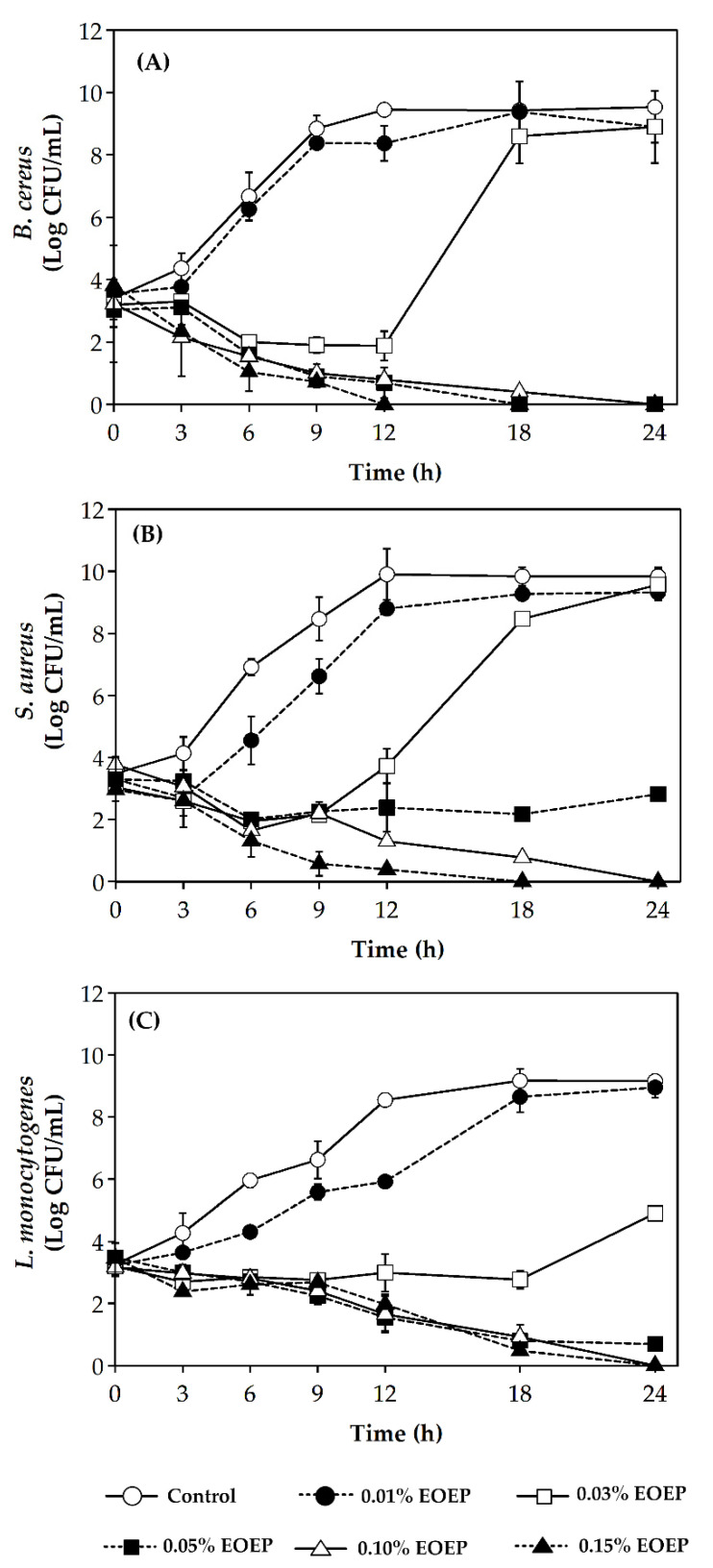
Killing curves for various concentrations of EOEP against *B. cereus* (**A**), *S. aureus* (**B**) and *L. monocytogenes* Scott A (**C**).

**Figure 4 molecules-25-03245-f004:**
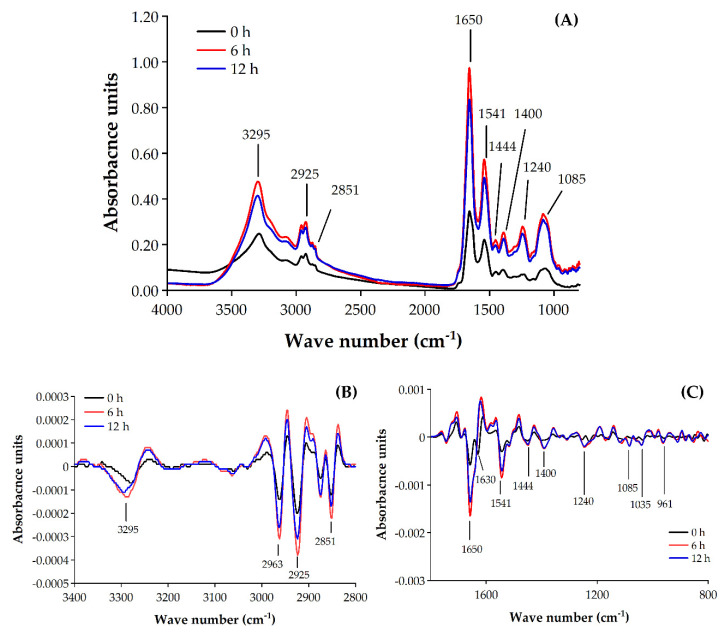
FTIR spectra region of *B. cereus* treated with 0.1% EOEP for 0, 6 and 12 h (**A**) and second derivative transformation spectra of *B. cereus* treated with 0.1% EOEP for 0, 6 and 12 h (**B**,**C**). Triplicate experiments were averaged.

**Figure 5 molecules-25-03245-f005:**
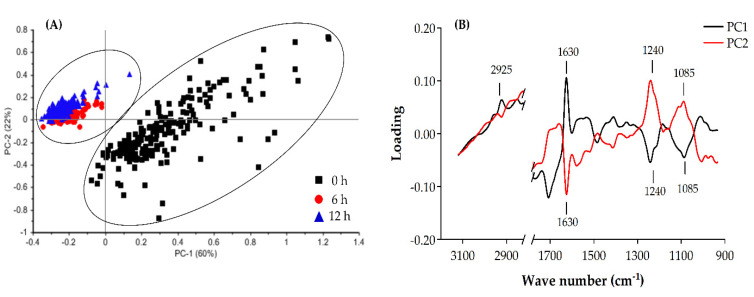
Scores plots (**A**) and loading plots (**B**) from PCA of the overall spectra data for *B. cereus* treated with 0.1% EOEP for 0, 6 and 12 h.

**Table 1 molecules-25-03245-t001:** Volatile compounds of EOEP analyzed by HS-SPME-GC/MS.

Compounds	LRI ^a^	% Area ^b^
**Monoterpenes (Total)**	1.55	
(−)-α-Pinene	944	0.12
Camphene	948	0.17
β-Myrcene	989	0.84
(+)-3-Carene	1014	0.07
α-Phellandrene	1014	0.01
*o*-Cymene	1034	0.01
*p*-Cymene	1034	0.01
d-Limonene	1039	0.15
β-Ocimene	1041	0.15
*δ*-Terpinene	1067	0.01
Bornyl acetate	1300	0.01
**Oxygenate monoterpenes (Total)**		0.40
Eucalyptol	1041	0.01
l-Fenchone	1101	0.01
Camphor	1163	0.12
Pinocarvone	1163	0.15
endo-Borneol	1184	0.04
Terpinen-4-ol	1193	0.03
α-Terpineol	1204	0.04
**Phenylpropanoids (Total)**	97.90	
Methyl chavicol (Estragole)	1210	19.36
*trans*-Anethole	1300	78.54
**Sesquiterpenes (Total)**	0.09	
α-Copaene	1402	0.01
Alloaromadendrene	1496	0.08
Total		99.94

^a^ Linear Retention Indices (LRI) on DB-5 MS capillary column. ^b^ Relative area (Percent of total peak area for each sample).

**Table 2 molecules-25-03245-t002:** The minimal inhibitory concentration (MIC) and minimal bactericidal concentration (MBC) values of EOEP against foodborne microorganisms determined by broth dilution method.

Tested Bacteria	MIC (% *v*/*v*)	MBC (% *v*/*v*)
**Gram-Positive Bacteria**		
*Bacillus cereus*	0.02	0.15
*Staphylococcus aureus*	0.03	0.15
*Listeria monocytogenes* 101	0.02	0.15
*Listeria monocytogenes* 108	0.02	0.10
*Listeria monocytogenes* V7	0.02	0.30
*Listeria monocytogenes* Scott A	0.02	0.10
**Gram-negative bacteria**		
*Escherichia coli*	5.00	5.00
*Pseudomonas aeruginosa*	>5.00	>5.00
*Salmonella* Typhimurium	>5.00	>5.00
**Lactic acid bacteria**		
*Lactobacillus plantarum* 541	>5.00	>5.00
*Lactobacillus plantarum* 844	>5.00	>5.00
*Pediococcus cerevisiae*	>5.00	>5.00
*Leuconostoc mesenteroides*	>5.00	>5.00

MIC = the lowest concentration of antimicrobial concentration required to inhibit the visible growth of bacteria tested in the test tube. MBC = the lowest concentration of antimicrobial concentration required to diminish at least 99.9% (3 Log reduction) of the initial bacterial amount at 24 h. The experiment was carried out in triplicate and the modal values are shown.

**Table 3 molecules-25-03245-t003:** Band area (× 10^−3^ cm^−1^) and band area changes of *B. cereus* treated with 0.1 % *v*/*v* of EOEP.

Wave Number (cm^−1^)	Possible Biomolecule Contributors	Contact Time			References
		0 h	6 h	12 h	
3295	Amid A in Proteins	246.37 ± 0.01b	475.86 ± 0.06a	414.12 ± 0.01a	[36,38]
2925	Fatty acids	159.49 ± 0.01c	299.66 ± 0.03a	274.81 ± 0.01b	[36,37,39]
2851	Fatty acids	107.39 ± 0.00b	190.44 ± 0.02a	177.77 ± 0.01a	[35,36,37]
1650	Proteinaceous content of cell	340.63 ± 0.02c	938.52 ± 0.10a	808.40 ± 0.05b	[32,37,40]
1630	Proteinaceous content of cell	272.01 ± 0.09c	609.91 ± 0.07a	514.76 ± 0.03b	[32,37,41]
1541	Proteinaceous content of cell	221.51 ± 0.01b	572.58 ± 0.06a	493.14 ± 0.03a	[37,41,42]
1444	Cell membrane (lipids, proteins)	79.61 ± 0.00b	211.66 ± 0.02a	189.53 ± 0.01a	[35,43]
1400	Cell membrane (amino acids, fatty acids)	89.56 ± 0.01b	244.34 ± 0.03a	215.35 ± 0.01a	[36,37,44]
1240	Phosphodiesters in nucleic acid	69.24 ± 0.01b	277.30 ± 0.03a	245.85 ± 0.01a	[37,45]
1085	DNA and RNA, phospholipids	92.20 ± 0.00b	334.54 ± 0.04a	308.73 ± 0.01a	[32,37,45]

Values are given as mean ± standard deviation. a–b Different letters in the same row indicate significant differences between the means obtained in Duncan’s test (*p* ≤ 0.05).

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
