# Peer review of "Assessment of Antimicrobial Activity, Mode of Action and Volatile Compounds of Etlingera pavieana Essential Oil"

_molecules, 2020, doi:10.3390/molecules25143245_

Round 1

Reviewer 1 Report

I reviewed the manuscript entitled Title: “Assessment of Antimicrobial Activity, Mode of Action and Volatile Compounds of Etlingera pavieana Essential Oil".

General comments:

This is an interesting research work related to the “Assessment of Antimicrobial Activity, Mode of Action and Volatile Compounds of Etlingera pavieana Essential Oil. The paper presents a comprehensive study described in a clear and detailed form in the results section and methodology. But the Discussion section is missing. The discussions are probably included in the results section but they are not clarified and they are also very poor. It is necessary to include the discussion section, or clarify if it is included in the results section. In the event that the discussions were in the results section, they are very limited and need to be further explored in relation to other studies with essential oils and their mode of action on the microorganisms tested.

Due to the omission of the Discussion section and the importance of this part and the poor discussion that could be included in the results section, this paper is Rejected.

Any way some questions are detailed below.

Line 94:

Pseudomonas aeruginosa (not abbreviated because there is the first time you mentioned)

Line 197:

Discussion section is missing.

 Line 237-248:

Why bacteria were cultured in a tube with MH and then were plated  in a different medium  (TSA)  and not in MHA?

Line 250:

You said that MIC is the lowest concentration that inhibit visible growth, but you don’t clarify where. Visible growth in the tube?

Line 255.

Since MIC and MBC  were carried out in MHB, why you used TSB when determined the Time Kill assay?

Reviewer 2 Report

To be honest I have no idea what was the aim. It seems authors obtained essential oil and instead of inject to GCMS made SPME only which of course doesn’t reflect the real composition. I have no idea why. SPME is used to make life easier as raw material is taken to do the measurement. So what is the thinking behind this procedure – I really don’t know and don’t understand. Also the paper is written in quite a complicated way as one need to read few times to get the point.

There is also no control – positive and negative in antibacterial results given. There is no discussion about which compounds can be responsible for activity. Why pure compounds were not tested.

The paper is really basic and the quality must be definitely improved. the identification of compounds must be checked

Some comments are included into the pdf file. Some other comments are as follow:

  • Section Plant material should be rewritten. In one sentence author repeat the name of the plant. Many information is missing. Who identified the plant/ what was the year of collection/ how did you dry the plant/ what about the voucher specimen? How much plant was taken to obtain essential oil? How much water was taken? Those are basic information!
  • The sentence “The percentage of the volatiles in the extract was 0.13% (v/w on fresh weight basis), which is slightly lower than the products obtained from other rhizome of Etlingera spp. which range between 0.28 and 0.45% (v/w on dry weight basis) [7]. Doesn’t have any sense. What does it mean volatiles in the extract? This you didntn even measure. You should deliver the content of EO obtained after hydrodistillation

Reviewer 3 Report

This manuscript deals with antimicrobial activity of volatile compounds of Etlingera pavieana. The authors had extracted the essential oils from this plant and analysed the chemical compounds on the basis of gas chromatography/mass spectrometry. They found that the main volatile constituents is trans-anethole. It is well known that trans-anethole shows antimicrobial activity, so the results of this study concerning the antimicrobial activity is not so attractive. However, they also elucidated the antimicrobial mechanism of volatile fraction using synchrotron FTIR analysis, and the results of this experiment is very interesting. I think this paper should be accepted with minor revisions as follows,

line 73

Please add the figure of chemical structures of main constituents such as trans-anehotle, estragole, and so on.

Table 2

Please add the explanations of MIC and MBC.

Reviewer 4 Report

The research described is interesting and useful but from the analytical point of view, some minor revision is required before acceptance for publication. The points which require further elaboration are:

  1. All standard deviations (S.D.) should be presented together with the number of measurement in order to have statistical meaning.
  2. Table 1: All numbers should be presented with the same significant figures. %Area is given with two decimals but the significant figures are different. Total Phenolic compounds is given as 97.893 but it should be 97.89.
  3. Table 2: ±0.00 means no random error which is not possible. The S.D. should be given with all zero digits and the first not zero and then the corresponding number should be given with the correct decimals. What is the meaning of >5 ± 0.00 ?? Most probably it is > 5.00?? ± 0.00? or just >5?
  4. Table 4: Requires attention, for example first row, 246±0.011 should be 246.00±0.01. Number of measurements??
  5. line 212: How the authors managed to weigh 5,000 ng? Is it after dilution of a stock solution??
  6. Is it FTIR Microspectroscopy or FTIR micro-spectroscopy (line 28)?

Round 2

Reviewer 1 Report

I think that they attended very well the questions and corrections.
the changes are significatives and I suggest the acceptance of the manuscript.

Author Response

Thank you very much for reviewing the revised manuscript and your kind response. The full description of EOEP was added, as shown in the attached file on Line 67-68.

"2.1. Yield Percentage and Analysis of the Volatile Compounds from the Rhizome of E. pavieana Essential Oils (EOEP)"

Reviewer 2 Report

Dear Editor

the manucript was already rejected by me due to the really serious issues

Author Response

Thank you for your response and for reviewing the first draft of the manuscript.

Regarding your comments, all issues were a point-by-point response to the comments, as the attached file previously. Many parts of the revised manuscript were rewritten, including a novelty and significance of the research and control sample which you may concern as serious issues.